# Digitally engaged physicians about the digital health transition

**Zsuzsa Győrffy**[1]*, **Nóra Radó**[1], **Bertalan Mesko**[2]

**1** Institute of Behavioural Sciences, Semmelweis University, Faculty of Medicine, Budapest, Hungary,
**2** Institute of Behavioural Sciences, Semmelweis University, Faculty of Medicine, Budapest, Hungary, The Medical Futurist Institute, Budapest, Hungary

* gyorffy.zsuzsa.mail@gmail.com

**Data Availability Statement:** In our country, studies in behavioural sciences not involving patients do not require ethical approval. This is a link to the section of the law in Hungarian (unfortunately, there is no English version). More information on Hungary: https://www.hsph.harvard.edu/region-map/research_project/

## Abstract

### Background

Digitalisation affects 90% of healthcare. Digital health, however, does not only refer to technological transformation but also has considerable cultural and social consequences. It fundamentally reshapes the roles of physicians and patients, as well as their relationship. Moreover, from the second half of the 20th century, the growing number of chronic patients and the increase in life expectancy have posed new challenges to the medical workforce.

### Objectives

To explore the digitally engaged physician's knowledge and attitudes towards digital health technologies and the transformation of the doctor-patient relationship.

### Methods

A qualitative interview study analysed with Interpretative Phenomenological Analysis (IPA). The study is based on qualitative, semi-structured interviews with 11 digitally engaged physicians from 9 countries. We identified four main themes emerging from e-physicians' responses and experience: 1) the past: intentions and experiences of change, 2) the present: the role of digital health and technology in the medical practice and their everyday challenges, 3) the present: the practical and ideal physician-patient relationship, and 4) the future: skills and competencies needed for working with e-patients and visions about the future of the medical practice.

### Results

The interviewed physicians state that digital health solutions could create a deeper doctor-patient relationship: knowledgeable patients are a huge help in the joint work effort and technology is the main tool for creating a more involved and responsible patient. Medical professionals in the future might rather get a role as a translator between technical data and the patient; as a guide in the jungle of digital health. However, the interviewed physicians also noted that digital transition today is more beneficial to patients than to their doctors.

hungary/ However this study was part of larger project so we have an ethics statement from Semmelweis University Ethical Committee (SE TUKEB: 268/2017) The participants of the qualitative study have given their official informed consent only regarding the use of quotes. As the study is completed no amendments can be made. Informed consents are available here: drive.google. com/drive/folders/1jWOl7EyJVYrcqpICjTJUTtjGIL-msVwY The data handling is regulated in Hungary by the Hungarian Healthcare Law 1997. https://net. jogtar.hu/getpdf?docid=99700154.TV&targetdate= &printTitle= Our priority is following these regulations. Data is available upon request. Contact for the long-term data access: Semmelweis University, Semmelweis University, Faculty of Medicine Institute of Behavioural Sciences. Contact person: Dr Agnes Zana Ph.D. Data protection officer zanagi72@gmail.com.

**Funding:** The author(s) received no specific funding for this work.

**Competing interests:** The authors have declared that no competing interests exist.

## Conclusions

We state that digitally engaged physicians are characterized by a kind of dichotomy: they use digital opportunities enthusiastically, but they also feel the difficulties related to digital health.

## Introduction

From the second half of the 20[th] century, there have been significant changes in the roles of physicians and patients, as well as the doctor-patient relationship [1]. The growing number of chronic patients and the increase in life expectancy pose new challenges to the medical work-force. In this transformation, the patient is no longer a passive sufferer but an active participant in the treatment process. The paternalistic model is replaced by a partnership, and the traditional hierarchy shifts towards a more patient-centered model [2]. Patient–centered medicine can be defined "as the medical practice aimed at improving the health outcomes of individual patients in everyday clinical practice, taking into account their preferences, objectives and values, as well as the available economic resources" [3]. Patients' rights have become fundamental: they require access to information and their medical records, and to be generally involved with the decisions made about them [4]. The latter is widely labelled in the literature as shared decision making. It has been defined as: 'an approach where clinicians and patients share the best available evidence when faced with the task of making decisions, and where patients are supported to consider options, to achieve informed preferences [5]."

This transition is facilitated by digitalization and the democratization of healthcare. The democratization of medical knowledge, access to data and dissemination of technologies creates a huge opportunity for patients [6]. These movements originate from a handful of countries such as the United States [7], Denmark [8], Australia [9], or New Zealand [10] where the related changes and concepts first appeared.

When personal computers became widely available in the 1990s, e-health emerged. When such computers could be connected into networks, telemedical services appeared. The rise of social media networks gave space to medicine 2.0 and health 2.0; while penetration of mobile phones and later smartphones summoned mobile health. A new phenomenon we call "digital health", and define as "*the cultural transformation of how disruptive technologies that provide digital and objective data accessible to both caregivers and patients leads to an equal level doctor-patient relationship with shared decision-making and the democratization of care*", initiated changes in providing care and practicing medicine. [11].

In light of the above, digital health does not only constitute a technological transformation; it also fundamentally reshapes the structure of healthcare systems from the doctor-patient relationship to treatment processes. Digital health is a paradigm shift based on the following changes: 1) the point of care is becoming the patient, not just the clinic or hospital, 2) medicine is more and more based on the individual rather than being general, and 3) it means partnership, data sharing and collaboration, instead of a hierarchy [11, 12]. Digital health initiated and later facilitated the e-patient movement. Empowered patients often have access to information, medical knowledge, and their own health data that was impossible to get before the digital era. It is critical for them to be in touch with fellow patients and to become highly active, engaged, equipped and educated in this process [13]. We could also summarize these changes using the bio-psycho-social–digital paradigm. The digital component covers the digital expansion of the biological self, the engagement of technology through smartphones, sensors, any

data gathering and/or medical devices, and the use of online networks. These factors are just as notable as the other biopsychosocial factors [14].

In our previous qualitative study, eleven e-patients, who are considered opinion leaders worldwide, shared their experiences and expectations about the future [15]. They explained that empathy, time, and attention are what e-patients need most from their physicians while using a myriad of advanced technologies in their care [16]. One of the key findings was that interviewed e-patients use communication tools and technologies such as search engines, social networking, social media, apps or sensors, but they did not mention any tool beyond the reach of an average consumer; therefore, being technology-savvy is a helpful but not principal requirement for empowerment. However, it turned out that e-patients require a doctor-patient relationship different from the traditional paternalistic model. They expressed how they need a partnership based on proper communication online and offline. Partnership also means that physicians and patients pay attention, to each other, articulate their needs clearly, and verbalize a common goal.

The above-mentioned circumstances have led to the e-physician phenomenon. We argue that patient empowerment, the spread of health technologies, the biopsychosocial-digital approach, and the disappearance of the very exclusive ivory tower of medicine result in a new role for physicians. Physicians will have knowledge of and positive attitude towards digital technologies, build up the doctor-patient relationship as a partnership and keep compassionate care as the fundamental basis of healthcare [17].

Our aim with this study is to explore the experience of digitally engaged physicians' knowledge and attitudes towards digital health technologies and the transformation of the doctor-patient relationship. We have conducted this research by making interviews with digitally engaged and empowered physicians, who have a special collaboration with their patients today and have knowledge about using digital health technologies for medical purposes. While the sample set is unique, we expected to gather insights that could help anticipate upcoming changes to the doctor-patient relationship and the future role of physicians in general.

## Methods

The methodological framework of our study is the same that was used by the authors in another study about the experience of e-patients published in BMJ Open [15]. The purpose of the common methodological background was to make the results of the two studies comparable. Ethical permission was obtained from the Ethical Committee of Semmelweis University, Budapest (No: 262/2017). Informed consent was obtained from participants prior to each interview.

### 1) Participants

The study is based on qualitative, semi-structured interviews with e-physicians. The purposive sampling method was used based on the following inclusion criteria: 1) active physician, 2) active online presence (social media, blogging, optionally telemedical services), 3) optionally having publications in medical journals, books or essays about their views or roles on healthcare; 4) or have spoken at medical conferences about digital health issues; and 5) speak English as a native or on nearly native level.

Looking at the geographical distribution of the potential interviewees, the participating countries were mainly chosen to resemble a developed state of a national digital health ecosystem, such as in the case of Australia, Canada, the United Kingdom, the Netherlands, France, Spain or the United States. All-in-all, we sent out 27 invitations by e-mail (3—Australia, 3—Canada, 2—France, 1—Hungary, 1—Israel, 2 Spain, 1—Netherlands, 1—Philippines, 2—UK, 11—US), and 13 participants accepted to be interviewed between February 1, 2019 and May 2,

**Table 1. The demographics of interviewees.** Written consents were obtained from every participant.

| Code/ name of participants | Gender | Age | Country of origin | Specialty | Type of Practice | Online presence |
|---|---|---|---|---|---|---|
| 1 | M | 54 | USA | Gastroenterology | Academic | Blog |
| 2 | F | 47 | The Philippines | Internal Medicine/ Endocrinology | Private Clinic | Social media / blog |
| 3 | M | 46 | Australia | Rheumatology | Private Group Practice | Social media, blog, editor for rheuma.com.au |
| 4 | M | 47 | USA | Gastroenterology | Gastroenterology practice | Social media |
| 5 | M | 36 | Australia | Physiotherapy | Private practice | Web / social media |
| 6 | M | 53 | USA | Surgery | Not-for-profit hospital | Social media |
| 7 | F | NA | New Zealand | Rheumatology | Public hospital | Social media |
| 8 | F | 35 | France | Pediatric emergency medicine | Public hospital | Social media/ website |
| 9 | M | 49 | Israel | Primary care | Clinic | Social media |
| 10 | M | 56 | USA | Internal Medicine | Primary care–private | Website / LinkedIn, Twitter |
| 11 | M | 41 | Hungary | ENT | Private and public hospital | Blog / Facebook |

2019 [18]. However, during the coding process, two interviews had to be dismissed. In one of the cases, the decision was made due to language, and thus interpretation problems, while in the other case, it turned out during the interview that the physician no longer met the criteria of being a practicing physician.

Table 1 shows the demographics and characteristics of the participants. When considering the response rate for our call to participate in the study, the usually lower response rate measured among doctors compared to the average response rate has to be mentioned as shown by Kellerman and Herold [19] or Cunningham et al. [20].

## 2) The content of the interviews

The same trained interviewer conducted all the interviews in a semi-structured manner using the following thematic blocks: the conversation started with personal questions about demographics and career choice, continued with queries about the potentially experienced changes since the beginning of the participant's profession, the functions of digital technologies, the relationship between physicians and patients, the experience with e-patients, challenges of medical education, and expectations and prospects for the future (See supplementary materials).

An interview guide was compiled and pilot tested among Hungarian physicians. Based on their initial feedback, the researchers (ZSGY & NR) modified the document. All interviews were conducted individually via a Skype call lasting between 60 and 90 minutes. A professional transcriber transferred the recordings to written form, and the interviewer checked the transcriptions for accuracy. Then, the final texts were sent back to each interviewee for confirmation.

After concluding the conversations, participants completed a short questionnaire about their personal (age, sex, nationality) and professional (years of practice, specialization) characteristics.

## 3) Analysis

Interpretative Phenomenological Analysis (IPA) was selected as the method for analysing the interviews conducted [21]. It is based on phenomenological and hermeneutic traditions and

represents an approach putting the individual and their interpretation of experience into the centre of attention. In such a way, the researcher is able to discern observations and treat the interviewed individuals as experts. The objective of the method is to explore and understand the experience in a highly detailed manner. The IPA approach leverages the idiographic method, which inspects in detail how individuals deal with situations of their lives.

Every one of the interviews was analysed separately by two researchers, who compared their results afterwards, to ensure validity. In each case, the analyses were conducted via the following steps: the inductive method was carried out for the primary coding. The two researchers (MB & GYZS) listened to the audio recordings of the interviews, as it is recommended by Rodham, Fox, and Doran [22]. Then, an initial line-by-line coding was performed on the verbatim transcribed interviews. This was augmented by initial comments and the selection of preliminary themes. The qualitative analysis was carried out with the help of the Atlas.ti 6.0. software.

As a next step, the codes and their keywords were used to delineate the most significant emerging themes. It was considered as such if it appeared in at least half of the interviews. Afterwards, these emerging themes were grouped together based on conceptual similarities and labelled with a descriptive title. The main themes were later broken down into various sub-themes, while outlining their linkages. Quotes from the interviews were assigned to the emerging themes, and the patterns of the themes in the individual interviews were attempted to be mapped out.

The two researchers discussed and developed both superordinate and subordinate themes to make sure the results, and evolved the final code structure, which was laid down in mutual agreement. Upon completing the qualitative analysis, the most significant outcome was sent back to the interviewees for feedback and comments for respondent validity. All physicians commented on the results, and the researchers built their recommendations into the final structure [23]. A list of superordinate themes and subordinate themes is demonstrated in Appendix 1.

## Results

We identified four main, superordinate themes emerging from e-physicians' responses and experiences (see Fig 1). We organized these themes along the timeline of a professional career: 1) the past: intentions and experiences of change, 2) the present: the role of digital health and technology in the medical practice and their everyday challenges, 3) the present: the practical and ideal physician-patient relationship, and 4) the future: skills and competencies needed for working with e-patients and visions about the future of a medical practice.

### Theme 1. The past: Intentions and experiences of change

**1.1. Career choice.** The 'warm-up' questions of the interview aimed at exploring the background of the interviewed physicians and map out their motivations for choosing the demanding profession of a physician and for dedicating their lives to it.

The interviewed physicians started to practice medicine between 1991 and 2004. Their career choice was mostly influenced by scientific curiosity and family motivation. One interviewed physician emphasized that his motivation was using science to allow him to help patients. Nine of them highlighted that they specifically wanted a profession where they could help and could be in touch with patients. One of them expressed their view in the following terms: *I decided that I really liked science and I really liked people and I certainly wanted the world to be a better place and doing medicine was the perfect combination of those three things.* [7].

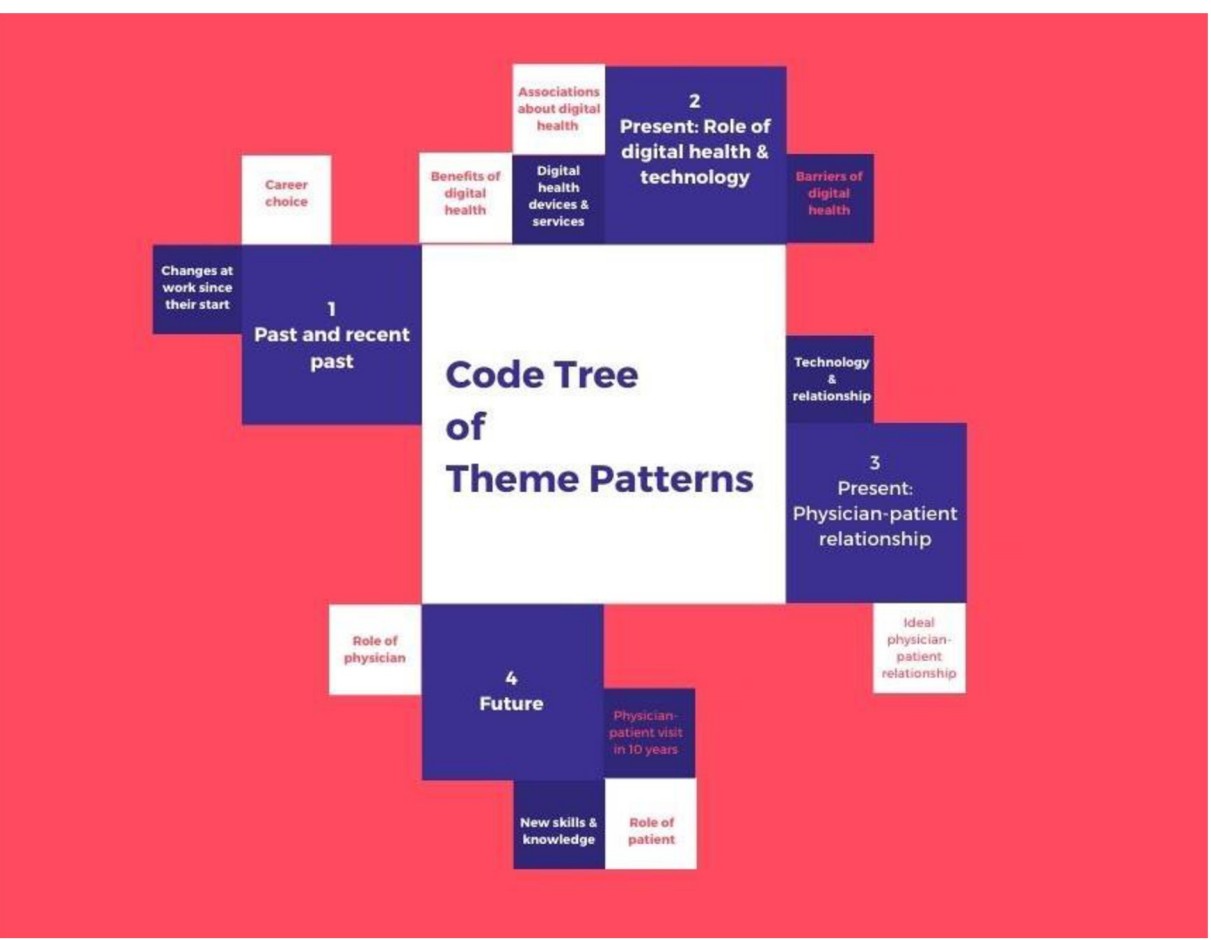

**Fig 1. Code tree of theme patterns.**

**1.2. Changes since the beginning of the profession.** The purpose of the second set of questions was to investigate whether the interviewees personally experienced any exogenous, structural changes regarding their professional life. The researchers were mainly interested in whether and how the technological revolution touched the everyday experiences of practicing physicians.

Interviewees have experienced different kinds of change since they have been practicing medicine. They mentioned changes related to information technologies and changes that new technologies facilitated in the doctor-patient relationship.

According to their views, making healthcare data digitized, the appearance of digital administration, the presence of telemedicine and the point-of-care not exclusively being the clinic or the hospital, but the patient's home have been crucial.

In one of the interviewee's words: '*I think probably the biggest changes involve technologically mediated communication between doctors and patients. I think that prior to the digital revolution, engagement between doctors and patients was strictly limited to face-to-face interactions. That has changed in that today we have a variety of technologically mediated platforms for connecting with patients. And so, the context of care between doctor and patient has completely evolved and changed.*' [1].

*Several interviewees reported that* their personal experiences made them realize what kind of changes have been taking shape and what consequences the digital transformation have. One of them said: '*I felt the "winds of change" in 2005 for the first time—when a patient knew a professional topic better than me because he/she got some information through the Internet. That's when I realized what a double-edged sword we have to deal with: on the one hand, a prepared patient is of tremendous help, on the other hand, re-calibrating wrong information means at least the same level of disadvantage.*' [11].

They emphasized that there are many more technological and pharmacological opportunities in treating patients. The appearance of the first smartphones in 2007 was a decisive turn in the evolution of the doctor-patient relationship. They emphasized that a growing number of patients started gathering information before meeting their physicians and were increasingly active in their treatment and decision-making. In the words of an interviewee:

'*Because of the Internet, social media and technology, my patients were coming to me with more information and they weren't looking to me to just solve a problem. They wanted to be involved in this problem.*'

[5].

Another one explained that '*the expectations from people in our community for their health care is much higher than they were when I first graduated over 20 years ago. So, when I first graduated people wanted some information but not a lot. And many people didn't really want much information at all. But now many people want a lot more information and the expectations for positive outcomes are much higher.*' [7].

They expressed how expectations towards physicians and the entire healthcare system have been growing due to patients' increased appetite for information. Among the changes, they also mention the issue of technologies used in the wrong way might lead to losing the human touch that encompasses the entire work of physicians. However, they emphasized that using the technologies in the wrong way is the root of the issue not technologies themselves.

## Theme 2. The present: The role of digital health and technology in the medical practice and their everyday challenges

**2.1. Associations with digital health.** When looking at associations with digital health and technology, we asked a question about what comes to physicians' mind when they hear the expression of digital health. We were curious about how digitally engaged doctors understand the term and whether there are any differences and/or common features when describing it. The first associations were related to new technologies and their consequences. They referred to those innovations that make it possible for medical professionals and patients alike to work together more efficiently. Every respondent said that digital health means a wide spectrum of possibilities from storing patient data digitally and having access to it anywhere to embracing robotic surgery, but also the phenomenon that the patient gets informed through online sources—already before turning to the doctor.

According to a physician's view, '*digital health is a broad term that is not a very helpful term. Because everything's digital now. We don't use the word digital banking. We just say banking, we don't say, you know, digital entertainment. We just say entertainment. So eventually digital health will just be health. But in the meantime, digital health broadly refers to the use of mainly remote monitoring technologies that allow extension of care beyond the hospital itself into the community through a variety of techniques including Bluetooth-enabled wearable bio sensors, smartphones, electronic health records, and its related technological infrastructure. And*

*it's to me a social and behavioural science about how to employ these technologies to improve patient outcomes.'* [4].

At the same time, they also attributed a cultural connotation to digital health as it facilitates patient empowerment. This feature becomes apparent in one of the interviewee's words as the following:

'*For me digital health is not about technology. Technology comes and technology goes. Technology is just the main tool for creating a more involved, responsible patient. The 'e' in e-Health doesn't stand for technology, it stands for the empowered patients.*'

[9].

**2.2. Experience with digital health devices and services.**   Closely related to the previous question, we asked physicians how well they know actual digital health applications, whether they use any such devices, applications, services, what do they use on a regular basis, and whether they already recommended anything to their patients. We think that the last question marks the highest stage of trust towards digital health.

For our question about the use of digital tools, every physician described how they have their own blog, how they use social media channels such as Facebook, Twitter, or YouTube and how they use electronic medical records. Half of interviewees mentioned using smartphone applications, telemedicine, and doing online consultation. One physician described using surgical robots and virtual and augmented reality applications in his medical practice. They also highlighted that the use of the different tools and online resources heavily depend on the patient. In one of the interviewee's words:

'*Some people want information but then it doesn't mean that they want to decide. They still need the doctor to decide for them. But there are some patients who are very empowered and feel that they want to decide for themselves. . .. You have to first explore what the patient wants or needs from us and not assume right away that they won't be taking our opinion to heart because otherwise they wouldn't have gone to the clinic to see us in the first place. That's the thing where we have to sort of say*, OK what does the patient want? *Does the patient want me to confirm what they had already searched and if it's applicable or are they coming here knowing that this is what they want done and we can't do anything about that anymore because they've made up their minds*?'

[2].

Interviewees suggest and recommend digital health tools such as smartphone applications, medical blogs, and websites to their patients. In the case of applications, they mentioned the question of responsibility and the importance of their personal experience in deciding what suggestion to make. In one of the interviewee's own words: '*I think we have the responsibility as providers and as teachers as educators to spread what's out there, so people get familiar with it and they start maybe demanding them and then maybe we accelerate the change that way.*' [6].

They again made it clear how different patients are and some patients find it important that they don't always have to come into the practice in person, others specifically ask for it. One physician described the situation as the following:

'*Patients love to not have to come into the office. It is not every patient. You know some patients are comfortable with the technology. Some patients prefer to come into the office for everything. That's OK. You know we have to meet patients where they are. We can't force our*

*patients to be digital. . . .We should encourage them to become engaged in their health and use whatever tools are appropriate and useful to them. But we must not ever force our patients to do that. We can't turn off our phone system because we have a patient portal.'*

[10].

**2.3. Benefits of digital health.**   In the next section of the interview, we asked physicians what benefits of digital health they perceive. Interviewees mentioned that they see the advantages of digital health especially in improving the access to care, making administration digital and the opportunities for further improvement that come with informed patients. Digital health tools also make the doctor-patient communication more intensive and resourceful. Opportunities for new communication channels lead to more equal communication between doctors and patients.

They mentioned how digital solutions help them collect information before the patient comes to see their caregiver. Patients can share their recent medical history on an application, they are already prepared, and physicians can then read it while the patient's in the waiting room. These solutions give them more time to engage with the patient, to talk to them face-to-face, and to understand what they really need to receive in the visit—this preparation might lead to a more efficient doctor-patient encounter. [4].

One of the respondent physicians already expressed a preference for such a changed relationship: '*I much prefer to work with a patient who is proactive about their health, who wants to have a conversation and work together.'* [5].

Other advantages included obtaining information real-time, getting feedback about their health, thus making real-time decision-making possible. The educated patient provides help in the process of treatment, although it is important to fight false information online. One of the interviewees explained the latter as the following:

'*I guess there is a perception attached to patients googling health symptoms and using social media, you know, that all this information is of poor quality, it's dangerous. Which, you know, we understand, and we need to be aware of.'*

[5].

However, they also articulated how through digital health and telemedicine, patients and physicians can save time and money for the healthcare system. According to one doctor: '*I would love technology to reduce inefficiencies, duplications, help to save money, prevent patients from getting unnecessary procedures, prevent people from wasting time. Can AI systems help me be a better clinician? Of course, they can. They'll give me better probabilistic abilities to make diagnosis, help me choose appropriate management and then hopefully I can then do the human aspects better.'* [3].

**2.4. Barriers of digital health.**   After vocalizing the benefits of digital health, we asked physicians about the barriers and drawbacks. Most of them described that digital health is more advantageous for patients today than for physicians. Among the barriers of the adoption of digital health technologies, they mentioned dealing with huge amounts of data and overregulation. Here is an example:

'*The challenge that our health system, certainly in the United States, is facing is that it is not fully prepared for the influx of data from patients. For example, if there are 10,000 patients in the practice of an internal medicine doctor in a small town and a doctor is getting. 10 or 12 of*

*these a day from the ten thousand patients, it creates a new kind of workflow for these doctors. Now it becomes their responsibility to help that patient understand which rhythms are truly abnormal or not abnormal. I think the software is going to get better in such a way that it will help patients better understand when there's a problem and when there's not a problem.'*

[1].

Another important question is the reimbursement, which manifests in different ways in different healthcare systems, but they mentioned that these tools are not yet an organic part of healthcare. *One of the physicians said*:

'*Many doctors are paid on what's called 'fee for service'. So, we only get paid when we interact with patients and consequently getting EKG readings all day by email isn't profitable.'*

[1].

They said if the systems are designed the right way and are validated clinically, they have a lot of potential but if not, they can be very damaging. More evidence based clinical trials and systematic reviews are needed. They mentioned how the lack of face-to-face interactions especially with telemedicine may cause problems such as misinformation and may lead to misdiagnosis. Technology when it is not used in the right way can influence the basic patient-physician relationship negatively. They also mentioned that electronic health records are not designed well. According to one of physicians a solution for that could be:

'*There is a company [..] developing a voice first interface that uses natural language processing that will listen to my conversation with a patient and then generate a note based on the passive listening to that conversation*'

A physician today focuses their attention on the computer instead of the patient. Therefore, systems need to be designed in a way that will facilitate the doctor in doing his job in a more convenient and efficient way.

Additionally, they brought up the issue of accessibility and how disparities in the access to technologies mean a widespread problem. They mentioned that access disparities of vulnerable populations result in accessing less health information online or communicating with providers using platforms such as patient portals.

**2.5. Questions of work-life balance.** The next group of questions referred to another important aspect of physicians' relationship with technology: to what extent and how it might impact not only their professional life, but also their work-life balance. All the interviewees addressed that digital health does not affect their work-life balance when good and strict regulations and rules are defined and discussed with patients. Such rules included how to communicate via e-mail and social media, what to do in case of an emergency and how often they respond to questions received via e-mails. *In the words of one physician*:

'*In order for the digital transformation to take place successfully, the way of thinking of both doctors and patients should be changed. We should accept that patients have the right to information, they should be considered as partners. But patients should also accept that the Internet and online communication aren't almighty either. For example, strict rules should be set up regarding in which period and to which problems we react. For example, I don't respond to an unknown patient's complaints at 10 p.m.'*

[11].

Many of the interviewees explained that they are not the only sources of information any more, therefore they do not worry about being contacted at inappropriate times. As explained by one interviewee, '*the truth is the patients actually have a lot more resources now than just doctors. Some of this stems from a real sense of self-importance that doctors have. That somehow patients are desperate to talk to us when in fact patients have a lot more resources at their fingertips. They have access to information, access to patient communities so much so that they don't necessarily need to talk to doctors quite as much.*' [1].

## Theme 3.: Present: The actual and ideal physician-patient relationship

**3.1. Relationship with patients and technologies.**   After describing their own relationship with digital technologies, we asked physicians to move onto an even more complex topic: their relationship with patients in the light of new technologies. We posed the question whether they think they might affect doctor-patient meetings, communication, and their relationship in general, and whether the influence they feel is in the more positive or more negative realm. Most of the interviewees described that they think digital health has a positive effect on the doctor-patient relationship. One of them said:

'*I think that the technology should empower this relationship rather than just take it away.*'

[8].

Some respondents explained how access to information has become democratized, and through that, how patients have become more informed and thus demand even more information. All these provide the opportunity for a more profound and thorough doctor-patient relationship. However, this today takes more time than before which puts pressure on the already overwhelmed healthcare system. There were opinions about how educating patients in a preventive manner could fill this time gap. As one of them explained:

'*I have some videos there that explain common conditions as well as posts and what I found fascinating was I would get some patients where it is as if the conversation started before they even met me because they watched a video. The conversation can actually become deeper because they come in armed with information and not like before where I had to explain everything. Now they know something, and we just have to reach a place where we sort of have a consensus as to how we're going to go about things.*'

[2].

They articulated how the decision-making process has been shifting towards patients and how the use of digital health technologies have recovered the balance between physicians and patients creating a partnership. This balance has not been in practice until digital health technologies emerged, as the paternalistic model had dominated for hundreds of years.

'*In participatory health, obviously, we talk about shared decision making and sort of working on the same footing and platform. I think being able to connect with patients digitally to share information, to share data has been a largely positive thing for me. The feedback I've received from patients even recently has been "It's really great!", "I love that we use these digital tools to connect and for me to be able to do my rehabilitation, to track my progress." and "My friends are jealous that we use this and my physician doesn't use this with me."*'

[5].

**3.2. The ideal physician-patient relationship.** '*Healthcare is not a spectator sport. It's a participatory sport.*' *[10]. This quote refers to how one of the interviewed physicians describes the ideal physician-patient relationship. Further elaborating on that, many of the respondents said that in* an ideal doctor-patient relationship, neither physicians nor patients are afraid of using technologies. And what's more important, patients and physicians facilitate each other's work through a shared learning process. As one of them described:

'*Patients and practitioners can actually work towards a common goal and use digital tools to not only communicate but to track information and in that way we can actually learn together. Much of this digital health allows us to collect data and information that we otherwise weren't able to have access to before. And if we can actually analyse this information together and make sense of it together, we put ourselves on the same page. I think that would be ideally what I would like to see.*'

[5].

Reaching the partnership necessitates patients to access information and their medical data. At the same time, physicians have a key role: they must suggest reliable resources, opportunities, and tools. In this partnership, the patient comes to the visit well prepared, is engaged with their care and has focused questions. Compared to their role today, patients are proactive in decision-making, prevention, and gaining information:

'*Ideally, the patient should be an active participant in the ongoing management of his or her care management. A prerequisite for this is access to one's health data, accompanied by eye-level explanations and proactive personalized prompts and digital tools that empower and encourage patients to be on top of their health condition from both prevention and ongoing disease management.*'

[9].

They mentioned enough time as a prerequisite: they can only maintain the partnership if they have enough time together; they understand and respect each other's decisions and goals. Active listening and empathy are also key in understanding the unique stories hiding behind data. It is important to point out that many e-physicians underlined how such partnerships are in fact team works involving family members and other healthcare professionals working on the best solution for the patient. One of them explained this as the following:

'*Well it really has to do with mutual respect, works in both directions. The doctor has to respect that the patient's viewpoint is what matters the most that they are an expert on their body and the symptoms that they're feeling. And the doctor is an expert on the physiology and the biology of the condition, and both have expertise, not just the doctor.*'

[4].

## Theme 4. Future: Skills and competencies needed for working with e-patients and visions about the future of a medical practice

The last set of questions was directed towards the future. We were curious to hear digitally engaged physicians' opinions and expectations about their future roles as physicians, what they expect from the patient of the future, what changes they expect in the next 5–10 years,

what implication they think this transformation has for the necessary skills and how they imagine a patient-doctor visit ten years from now.

**4.1. The future role of physicians.** All interviewees mentioned that technology will play a much bigger role in their job and will be inevitable. They cannot see properly what other changes it will also initiate, such as handling data, cybersecurity, or ethical issues. However, they expressed how they believe that as practicing medicine encompasses life-long learning, physicians also have to learn to integrate new technologies from artificial intelligence and virtual reality to telemedicine into their job and into educating patients better. In their opinion as discerned from their responses, the physician must become a sort of Swiss army knife who is able to handle complex problems; helps patients navigate in the healthcare jungle; facilitates decision-making while communicating with other members of the healthcare team. A medical professional is going to be a mix between health professional, IT professional, data scientist all in one and they imagined medical care to move outside the clinic into the community. As one of them described:

'*A health professional in the future is going to be expected to be a bit of a kind of like a Swiss army knife, someone who has multiple skills and knowledge sets. Again, this idea of health professional who understands digital health is not going to be such a separate thing. I think our idea of what a medical professional is going to be is a mix between health professional, IT professional, data scientist all in one. I think medical care is going to move more and more outside the clinic into the community.*'

[5].

Being a caregiver will mean being a mediator helping patients orientate among heaps of unverified information, opportunities, and data, according to the respondents. One of them explained:

'*I would see myself as someone who is a docent or someone who mediates all this information that's coming in and helps them make decisions in only a way that another human could do.*'

[1].

This also means the increased responsibility of patients for their own health. In the words of one physician:

'*They need to understand that the main responsibility for retaining and improving their health is on their shoulders, it is not on my shoulders as a physician. I need to provide the tools and the guidance, but weight is on their shoulders.*'

[10].

According to their opinion, technology is the new hope that provides the chance for freeing physicians from time-consuming, repetitive, and administrative tasks. Thus, in 10 years, their job could be more interesting, efficient, and exciting. They will have more time and attention for their patients; and empathy, attention and the human touch will return to their daily jobs. These changes require physicians to be proactive instead of reactive, as it is often the case today.

**4.2. New skills and knowledge needed.** When asked about the implications of new technologies for doctors' talents, they said how physicians' relation to technologies need to change by also accumulating new skills through an improved medical curriculum. The skill of filtering

information might become more important than lexical knowledge. Also, medical professionals of the future should be able to embrace the cultural changes that come with digital health. As one of them articulated:

'*they should know how to accept that the patient has exactly the same information as they do. And both parties should know how to manage the possibilities created by information communication.*'

[11].

Predictive and preventive approaches; social skills; emotional intelligence and new communications skills will also be crucial. Additionally, healthcare organizations need to help their members know more about all these new technologies, they have to provide these technologies to the members and also must motivate their clinicians to get out there and be ahead of these technologies so that they understand them and can embed them in their practice. In this process it is important being proactive rather than reactive.

**4.3. The future role of patients.**   Digitally engaged physicians articulated that they believe patients of the near future will be technology-savvy, more engaged with their care, and will manage prevention and health better. Patients of the future are already here in the form of e-patients. In one of the physician's words:

'*I think that the patient of the future is here today in some ways which is that some patients are engaged. They want to engage in their health and want to do things to help their health. And they want to shop around. They want information they want to find out where the best places to do treatment for their cancer or get this test done or so on. They want to engage their physicians in conversations and discussions about different options. They want shared decision making. And I think that you know the patients will increasingly seek out doctors who will do this with them rather than just accepting the doctor that they have.*'

[10].

In the future, physicians expect to have much more data about their patients making it possible to provide truly personalized care. Patients will be educated, informed, and will have a better picture about their health or disease management than today. Interviewees also assumed that the doctor-patient relationship will never become purely online, however, the role of the physicians will be closer to a guide providing support.

**4.4. The physician-patient visit in 10 years.**   All interviewed physicians think that the doctor-patient relationship will mainly take place in person. At the same time, they expect the visit to be immersed with advanced technologies from augmented and virtual reality to artificial intelligence and health sensors. They see advances in precision medicine, personalized medicine, the availability of digital tools and patients' abilities to self-manage, tools automating workflows and work processes. They would like to incorporate remote consultation, self-tracking, quantification, analytics, genomics in their practices. But they mentioned that the real physician-patient visit will always be inevitable. Two of them explained all this as the followings:

'*There'll be change but not that sweeping. I'm sure the virtual patient-doctor meeting will be more widespread, but nothing will replace the personal connection/exam. But I'm sure that it will be paperless—with every piece of data accessible virtually anywhere, anytime. But I'm*

*sure that the personal, real patient-doctor meeting will remain indispensable also 10 years from now.'*

[11].

'*Another thing I would love during a consultation is the use of voice recognition and simple AI so that I don't have to type. So, I don't have to move the mouse. The computer listens to me, it transcribes the consultation and then I just edit it. If I say Alexa please print a script for this and this, it happens without me having to write. This would be very nice. And then I can spend my full attention on the patient. I can talk to the patient, I can look in their eyes, examine them and not spend 25–50 percent of the consultation tapping on the computer while trying to engage with the patient.'*

[3].

The main change they agreed upon was the notion of getting a clearer picture of the patient even before the visit through the constant influx of data and information. They also mentioned how monitoring could become more efficient through that.

'*Without a doubt in the next 5 to 10 years we'll be using a lot more technology but how and when, you know, no one can predict obviously. I think as a physician I will be very much still a physician. I may be using AI and a computer, digital health to help me be a more effective physician. And hopefully more of the drudgery and the irritating work can be given to a machine counterpart and I can spend time doing the things I like and giving value in a better way rather than doing things that can be done by AI.'*

[3].

## Discussion

This study attempts to describe digitally engaged physicians' attitude and knowledge related to digital health issues and tries to explore their relationships with their patients. Their vision of the near future around changes in medicine within the next 10 years is also a part of our investigations.

Participants mentioned that the most important change since they started to practice medicine has been the increase in technological possibilities for treating patients. These changes are not only digital, but also impact the basics of the doctor-patient relationship. In fact, these have had a serious impact on all the fields of medicine. Participants emphasized that the Internet and the spread of smartphones have fundamentally transformed the physician-patient relationship: patients have access to more information and data—essentially the same as their physicians. These new channels of communication are shifting the doctor-patient relationship and providing patients with more opportunities to engage [24]. Patients are coming to their caregivers with more information and they want to be involved in every step of the treatment process. Similarly to the results of our e-patients study, digitally engaged physicians do not use special technologies the average physicians don't have access to. They just use social media, smartphones, and online consultation [15]. For separating their professional and private lives, physicians use well-designed techniques that optimize their time management. They mentioned that strict rules should be set up regarding when they react to patients' messages and to what kind of problems. This should be implemented for other physicians to avoid burnout in their professional life.

With digital health technologies, the patient is gradually becoming the point-of-care; diagnostics and treatments can be offered from a distance; patients can be involved in their care to release the burden on medical professionals; To maximize the benefits of modern digital technologies in improving patient outcomes, the NHS in the United Kingdom expressed the need for clinicians to use e-patients' expertise to improve healthcare and to re-design care using digital health [25]. The "bottom-up medicine" phenomenon has appeared, in which digitally empowered patients will take charge of their own healthcare [26]. According to the AMA Digital Health survey there has been an increase in the number of physicians that see a definite advantage in digital tools, and adoption of digital tools has grown significantly among all physicians regardless of gender, specialty or age [27]. Similarly to our findings the most important benefits for them were the improvement of efficiency and the increase in patient safety, and diagnostic ability.

The interviewed physicians' stated that digital health solutions could create a deeper doctor-patient relationship: they believe that knowledgeable patients are a huge help in the joint work effort and technology can become the main tool for creating a more involved and responsible patient. Digital health is a tool, which can improve communication in care and, if supported by social media, it can also induce more trust from the patients and make them more engaged in their own care. There are, however, numerous concerns when it comes to security, efficiency, and misinterpreting information and data [28–30].

Our findings directly reflect the Mutual Participation Model of the physician-patient relationship, in which the patient is an expert and plays a crucial role in the treatment process [31, 32]. Digital health opens communication in healthcare [13]. In a systematic review on patients' online health information seeking and its influence on the doctor-physician relationship, the authors found that health seeking behaviour can improve that depending on whether the patient discusses the information with their physician and the nature of their prior relationship [33].

Physicians mentioned that there are many advantages to digitalization, but they also could see the potential drawbacks. According to them, digital health today is more beneficial to patients than to their doctors, mostly due to the questions of financing, valid tools, evidence-based studies, data management, and accessibility issues. The scoping review of de Grood et al cost and liability issues, unwillingness to use e-health technology, and training and support were the most frequently mentioned barriers and facilitators to the adoption of e-health technology [34]. The results showed that there is uncertainty about the costs, implementation strategies, and the impacts of digital health technology.

Every second physician is involved in burnout according to a study of the AMA, and several studies have highlighted the same observation worldwide. It has been found that unmanageable technological challenges are among the major causes of burnout [35]. Adoption and implementation of new technologies require new, special skills from medical professionals. Health information technology (HIT) today is, similarly to the difficulties to make work-life balance work, a stress factor. The Electronic Health Record (EHR) and the computerization in the practices may have potential benefits as well: to facilitate the collaborative physician–patient relationships, opportunity to share online medical information and treatment plan decisions among others [36].

A 2016 study has found that doctors spend almost half of their working day on desk work and less than a third face-to-face with their patients [37]. A constantly growing number of studies have concluded a connection between reduced satisfaction in work and the increasing amount of digital administration [38]. What we argue is that digital health solutions and digital administration are two hugely different concepts: effective digital health tools aim to deliver an easier way of coping with the administrative tasks, among other things.

The greatest novelty of a medical visit in the near future is expected to be the preventive approach: by monitoring the available data, doctors will be able to see potential issues much earlier and get to know the health of their patients without them arriving at the office. Technology is the new hope that will save doctors' time and energy; and reduce the repetitive and administrative work. Some tasks are just much easier to be done using digital health solutions, such as artificial intelligence or voice-to-text interfaces. These have the potential to enhance diagnostic accuracy to such a level that doctors will finally be able to focus on the relationship with their patients and spend quality time with them [39].

According to the visions they shared, in around 10 years, medical work will be much more interesting, exciting, and productive. Patients will receive more time, attention and empathy and the human touch will return. The development of technology and information filtering capabilities will be essential. The physicians also emphasized the importance of "behavioural skills": social skills, emotional intelligence and new forms of communicative skills will be needed and understanding the role of social determinants and the ability of critical thinking will also be crucial.

## Strengths and limitations

The main strength of this study is being the first to explore the personal experiences and attitudes of digitally engaged physicians who are experienced worldwide about digital health technologies, and their current and future relationship with their patients. To the best of our knowledge, this is the very first qualitative study on the topic of the transition of the doctor-patient relationship from the point of view of digitally engaged physicians.

Our method allowed us to gain in-depth knowledge of underlying reasons and motivations of e-physicians for adopting digital health and for working with e-patients.

The limitations of the qualitative method (IPA) include that the results cannot be generalized because of the small cohort size and the way participants were chosen. Further limitation is that only digitally engaged physicians speaking English were included in the study.

Our qualitative approach was basically aimed at e-physicians, who use digital technologies in their daily practice, who have a scientific / popular background in this field, and who also support this process with professional publications and online presence. So, this is a very special study population therefore our setting and results can only be interpreted for this sample.

## Conclusions

The role of the doctor is in transition: today already and in the near future, doctors will have to perform more complex tasks: health IT, helping in the digital orientation of patients, and filtering information for them, among others. They imagine themselves as a sort of a guide, transforming into a mediator based on efficient communication with the patient. Pertaining to the concept of apomedication [40], the researchers found that digitally engaged physicians consider themselves as a sort of guide, who undertake a "guardian" and information managing function in the description, collection, and sharing of credible content in the online space [33].

We conclude that, for a successful leap from the hierarchical patterns towards the 21st century doctor-patient relationship, the future generation of physicians should be trained differently and should be prepared for all the above-described changes. Medical school curricula should emphasize health and prevention rather than only diseases and pathology via the newest digital technological solutions and medical students have to prepare for the predictive and proactive working environments, including their new role as a guide or mediator for the patient and their use of digital health technologies [41, 42].

## Supporting information

**S1 File.**
(DOCX)

## Acknowledgments

The authors would like to express their gratitude to the participating e-physicians.

## Author Contributions

**Conceptualization:** Zsuzsa Győrffy, Nóra Radó.

**Investigation:** Nóra Radó.

**Writing – original draft:** Zsuzsa Győrffy.

**Writing – review & editing:** Zsuzsa Győrffy, Nóra Radó, Bertalan Mesko.

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
