## [Decision Letter · Decision Letter 0]

12 Jun 2020

PONE-D-20-11916

Physicians’ role in the digital health transition: A qualitative study among digitally engaged physicians

PLOS ONE

Dear Dr. Győrffy,

Thank you for submitting your manuscript to PLOS ONE. After careful consideration, we feel that it has merit but does not fully meet PLOS ONE’s publication criteria as it currently stands. Therefore, we invite you to submit a revised version of the manuscript that addresses the points raised during the review process.

Two Reviewers have evaluated the manuscript. They found merit in the manuscript but also suggested revisions in terms of completeness of methodological information, background and conclusion. 

We look forward to receiving your revised manuscript.

Kind regards,

Stefano Triberti, Ph.D.

Academic Editor

PLOS ONE

Journal Requirements:

4. Your ethics statement must appear in the Methods section of your manuscript. If your ethics statement is written in any section besides the Methods, please move it to the Methods section and delete it from any other section. Please also ensure that your ethics statement is included in your manuscript, as the ethics section of your online submission will not be published alongside your manuscript.

5. Please include a caption for figure 1.

Reviewers' comments:

Reviewer's Responses to Questions

**Comments to the Author**

1. Is the manuscript technically sound, and do the data support the conclusions?

Reviewer #1: Yes

Reviewer #2: Yes

2. Has the statistical analysis been performed appropriately and rigorously? 

Reviewer #1: N/A

Reviewer #2: N/A

3. Have the authors made all data underlying the findings in their manuscript fully available?

Reviewer #1: Yes

Reviewer #2: Yes

4. Is the manuscript presented in an intelligible fashion and written in standard English?

Reviewer #1: No

Reviewer #2: Yes

5. Review Comments to the Author

Reviewer #1: The paper is interesting and makes a contribution to the literature on this topic. The quotes are well-selected. The methods step of going back to the participants to validate their findings (member checking) is also good.

I have a number of important suggestions, however.

The title and running head should be improved.

Statements are occasionally written in too-global and generalizing a fashion. For example, at the bottom of p3, the words "reported that" should be included to qualify the findings: "what opinion leader reported that e-patients..."

Similarly, when reporting all findings, please remind the reader each time that participants "reported" or "said" or "commented". Otherwise, the statements are reported as if true, and the statements are likely not necessarily true for anyone else but those who were interviewed.

It is preferable to write themes as full sentences rather than as topic phrases. As phrases, they do not convey sufficient information. Instead of, for example: The past: intentions and experiences of change, the theme should preferably be worded something like: Physicians described how their initial intentions changed over time.

Another example: instead of: Benefits of digital health, it would be better worded as something like: Physician informants described benefits from digital health to improve care.

It is also preferable to insert text between quotes for improved coherence and readability. A sentence or two following each theme is also desirable to help sum up the theme and transition to the next theme.

It would be helpful to include suggestions on p12 following the paragraph that electronic health records are not designed well. What would help?

Additional references are needed in places, such as, p14, first sentence; beginning of 3.2; the first paragraph of 4.1; and other places in the findings and in the discussion where assertions are made that need references to support them.

References that also should be added include:

Shield RR, Goldman RE, Anthony DA, Wang N, Doyle RJ, Borkan J. Gradual electronic health record implementation: new insights on physician and patient adaptation. Ann Fam Med. 2010;8(4):316‐326. doi:10.1370/afm.1136

Doyle RJ, Wang N, Anthony D, Borkan J, Shield RR, Goldman RE. Fam Pract. 2012 Oct;29(5):601-8. doi: 10.1093/fampra/cms015. Epub 2012 Feb 29.PMID: 22379185

The methods should be described in more detail with steps more thoroughly delineated. Some questions to answer are:

how many participants were asked to be included in the study?

what questions were asked

what were the codes

the code tree figure should be explained in the text

The low response rate should be additionally noted in the discussion under limitations

The discussion is too wordy and is often repetitive. The findings should be explored in more depth instead.

Limitations are not sufficient and should include the small sample size.

A conclusion should be added!

The English is mostly good but needs proofreading and is awkward in numerous places. It needs editorial assistance.

Thank you for the opportunity to review this paper.

Reviewer #2: The article presents the results of a qualitative study that explore the digitally engaged physician’s knowledge and attitudes towards digital health technologies and the transformation of the doctor-patient relationship.

The manuscript is well organized ad well written. I think the most lacking part is the introduction for which I have minor revisions. As far as the methodology and results are concerned, I think they are well written, clear and comprehensive. So, I believe this article should be considered for publication.

Regarding the introduction, I think some concepts can be better integrated and expanded.

-For example, the concept of “patient-centred medicine model”. The authors may be inspired by some of the following articles:

Mezzich, J., Snaedal, J., Van Weel, C., & Heath, I. (2010). Toward person-centered medicine: From disease to patient to person. Mount Sinai Journal of Medicine: A Journal of Translational and Personalized Medicine, 77(3), 304-306.

Sacristán, J. A. (2013). Patient-centered medicine and patient-oriented research: Improving health outcomes for individual patients. BMC Medical Informatics and Decision Making, 13(1), 6.

-Authors could explain the concept of shared decision making: the process. Of decision making that involves both sides of the patient and doctor in making decisions regarding, for example, therapies. Regarding this concept, authors could take into consideration the following articles:

Barry, M. J., & Edgman-Levitan, S. (2012). Shared decision making—The pinnacle patient-centered care.

Glanz, K., Rimer, B. K., & Viswanath, K. (Eds.). (2008). Health behavior and health education: theory, research, and practice. John Wiley & Sons.

Renzi, C., Riva, S., Masiero, M., & Pravettoni, G. (2016). The choice dilemma in chronic hematological conditions: why choosing is not only a medical issue? A psycho-cognitive perspective. Critical reviews in oncology/hematology, 99, 134-140.

Frosch, D. L., & Kaplan, R. M. (1999). Shared decision making in clinical medicine: past research and future directions. American journal of preventive medicine, 17(4), 285-294.

Gorini, A., & Pravettoni, G. (2011). An overview on cognitive aspects implicated in medical decisions. European journal of internal medicine, 22(6), 547-553.

- Also, the concept of digital health requires more explanation as it is a fundamental theme of the paper. The authors could explain and described the different types of digital health that are currently in use, for example: telemedicine, eHealth/mHealth, Digital Therapeutics etc.

Regarding the methodology, in the paragraph “the content of interview”, I think that inserting some examples of questions used in the interview can be useful for readers to better understand.

There is an error at the top of page 20, please change “and accessibility issuesThe scoping review…” with “and accessibility issues. The scoping review…”.

Finally, regarding the discussion, I think this mainly focuses on the advantages of using technology for patients, leaving the advantages for doctors, who are the protagonists of the article, somewhat in the background. Authors should emphasize the reasons why technology can be useful for the doctor; there are a lot of literature on the benefits of technology for patient centered medicine.

For example, the use of technology can help save time during interviews, also allowing the doctor to focus more on what the patient's real problem and concern is which may not concern the mere elimination of the symptom and, at times, not even the treatment of the disorder.

6. PLOS authors have the option to publish the peer review history of their article (what does this mean?). If published, this will include your full peer review and any attached files.

Reviewer #1: No

Reviewer #2: No

---

## [Author Response · Author response to Decision Letter 0]

13 Aug 2020

We are very grateful to both reviewers for all the valuable comments. We put our responses after the raised questions or comments.

Reviewer #1: The paper is interesting and makes a contribution to the literature on this topic. The quotes are well-selected. The methods step of going back to the participants to validate their findings (member checking) is also good.

I have a number of important suggestions, however.

Thank you for your time and effort and thorough feedback We have tried to do our best to implement all requested and necessary changes.

The title and running head should be improved.

 We attempted to clarify and reorganize both the title and the running head:

Title: Digitally engaged physicians about the digital health transition

Short title: A qualitative study on digitally engaged physicians

Statements are occasionally written in too-global and generalizing a fashion. For example, at the bottom of p3, the words "reported that" should be included to qualify the findings: "what opinion leader reported that e-patients..." 

Thank you for the comment, we have specified these sentences.

Similarly, when reporting all findings, please remind the reader each time that participants "reported" or "said" or "commented". Otherwise, the statements are reported as if true, and the statements are likely not necessarily true for anyone else but those who were interviewed –

We believe that specifying statements to separate the quotes from our interviewees and the research description to be sufficient in the paper.

It is preferable to write themes as full sentences rather than as topic phrases. As phrases, they do not convey sufficient information. Instead of, for example: The past: intentions and experiences of change, the theme should preferably be worded something like: Physicians described how their initial intentions changed over time.

Another example: instead of: Benefits of digital health, it would be better worded as something like: Physician informants described benefits from digital health to improve care.

Thank you for the comment. We address this requirement but we would like to point out that similar articles in the field tend to use short headings: https://journals.plos.org/plosone/article?id=10.1371/journal.pone.0191635

The reason for short headings is to make the study better searchable.

It is also preferable to insert text between quotes for improved coherence and readability. A sentence or two following each theme is also desirable to help sum up the theme and transition to the next theme.

We are grateful for the reviewer for making these suggestions. We have improved the results and added more relevant sentences.We hope that now we sum up all of the themes in the manuscript.

It would be helpful to include suggestions on p12 following the paragraph that electronic health records are not designed well. What would help?

Thank you. The manuscript has been amended with a potential solution for that by one of the interviewed physicians.

Additional references are needed in places, such as, p14, first sentence; beginning of 3.2; the first paragraph of 4.1; and other places in the findings and in the discussion where assertions are made that need references to support them. később magyarázom meg.

Thank you for the important remark: We attempted to explain these in the discussion part.

References that also should be added include:

Shield RR, Goldman RE, Anthony DA, Wang N, Doyle RJ, Borkan J. Gradual electronic health record implementation: new insights on physician and patient adaptation. Ann Fam Med. 2010;8(4):316‐326. doi:10.1370/afm.1136

Doyle RJ, Wang N, Anthony D, Borkan J, Shield RR, Goldman RE. Fam Pract. 2012 Oct;29(5):601-8. doi: 10.1093/fampra/cms015. Epub 2012 Feb 29.PMID: 22379185

We implemented these references into the manuscript.

The methods should be described in more detail with steps more thoroughly delineated. Some questions to answer are:

how many participants were asked to be included in the study?

We sent out 27 invitations by e-mail (3 - Australia, 3 - Canada, 2 - France, 1 - Hungary, 1 - Israel, 2 Spain, 1 - Netherlands, 1 - Philippines, 2 - UK, 11 - US), and 13 participants accepted to be interviewed between February 1, 2019 and May 2, 2019. However, during the coding process, two interviews had to be dismissed. In one of the cases, the decision was made due to language, and thus interpretation problems, while in the other case, it turned out during the interview that the physician no longer met the criteria of being a practicing physician. So our sample contains 11 physicians.

what questions were asked

The questions are available in the supplementary materials attached.

what were the codes

Please find the superordinate and subordinate codes below:

1. Past and recent past 1.1. career choice

1.2. changes at work since they’ve been on their career

2. Present : Role of digital health and technology 2.1. Digital health associations

2.2. Digital health devices/services

2.3. Benefits of digital health

2.4. Barriers of digital health

3. Present: physician-patient relationship 3.1. Technology and relationship

3.2. Ideal physician-patient relationship

4. Future 4.1. Role of physician

4.2. New skills and knowledge

4.3. Role of patient

4.4. Physician-patient visit in 10 years

the code tree figure should be explained in the text .

Thank you for the comment, we explained the containts of the code tree in the results part in detail.

The low response rate should be additionally noted in the discussion under limitations

We added this limitation in the discussion part.

The discussion is too wordy and is often repetitive. The findings should be explored in more depth instead. 

Thank you for this comment. We attempted to shorten the discussion part and higlighted the most important findings.

Limitations are not sufficient and should include the small sample size.

Thank you for the important comment. Our qualitative approach was basically aimed at e-physicians, who use digital technologies in their daily practice, who have a scientific / popular background in this field, and who also support this process with professional publications and online presence. So, this is a very special study population therefore our setting and results can only be interpreted for this sample. We put this limitation in the manuscript.

A conclusion should be added!

We are grateful for the reviewer for making this notice. We attached the conclusion part.

The English is mostly good but needs proofreading and is awkward in numerous places. It needs editorial assistance.

We carefully proofread the manuscript.

Thank you for the opportunity to review this paper.

Reviewer #2: The article presents the results of a qualitative study that explore the digitally engaged physician’s knowledge and attitudes towards digital health technologies and the transformation of the doctor-patient relationship.

The manuscript is well organized ad well written. I think the most lacking part is the introduction for which I have minor revisions. As far as the methodology and results are concerned, I think they are well written, clear and comprehensive. So, I believe this article should be considered for publication.

Regarding the introduction, I think some concepts can be better integrated and expanded.

Thank you for your useful comments we have taken your advice and improve the manuscript.

-For example, the concept of “patient-centred medicine model”. The authors may be inspired by some of the following articles:

Mezzich, J., Snaedal, J., Van Weel, C., & Heath, I. (2010). Toward person-centered medicine: From disease to patient to person. Mount Sinai Journal of Medicine: A Journal of Translational and Personalized Medicine, 77(3), 304-306.

Sacristán, J. A. (2013). Patient-centered medicine and patient-oriented research: Improving health outcomes for individual patients. BMC Medical Informatics and Decision Making, 13(1), 6.

Thank you for the comments: we explained the concept of the „patient centered-model” in the text.

-Authors could explain the concept of shared decision making: the process. Of decision making that involves both sides of the patient and doctor in making decisions regarding, for example, therapies. Regarding this concept, authors could take into consideration the following articles:

Barry, M. J., & Edgman-Levitan, S. (2012). Shared decision making—The pinnacle patient-centered care.

Glanz, K., Rimer, B. K., & Viswanath, K. (Eds.). (2008). Health behavior and health education: theory, research, and practice. John Wiley & Sons.

Renzi, C., Riva, S., Masiero, M., & Pravettoni, G. (2016). The choice dilemma in chronic hematological conditions: why choosing is not only a medical issue? A psycho-cognitive perspective. Critical reviews in oncology/hematology, 99, 134-140.

Frosch, D. L., & Kaplan, R. M. (1999). Shared decision making in clinical medicine: past research and future directions. American journal of preventive medicine, 17(4), 285-294.

Gorini, A., & Pravettoni, G. (2011). An overview on cognitive aspects implicated in medical decisions. European journal of internal medicine, 22(6), 547-553.

- Also, the concept of digital health requires more explanation as it is a fundamental theme of the paper. The authors could explain and described the different types of digital health that are currently in use, for example: telemedicine, eHealth/mHealth, Digital Therapeutics etc.

Thank you for the suggestions. We explained and added both terms to the background section.

Regarding the methodology, in the paragraph “the content of interview”, I think that inserting some examples of questions used in the interview can be useful for readers to better understand.

We added our questions as supplementary material and quoted them directly in the results part as well.

There is an error at the top of page 20, please change “and accessibility issuesThe scoping review…” with “and accessibility issues. The scoping review…”.

Thank you, we changed it.

Finally, regarding the discussion, I think this mainly focuses on the advantages of using technology for patients, leaving the advantages for doctors, who are the protagonists of the article, somewhat in the background. Authors should emphasize the reasons why technology can be useful for the doctor; there are a lot of literature on the benefits of technology for patient centered medicine.

For example, the use of technology can help save time during interviews, also allowing the doctor to focus more on what the patient's real problem and concern is which may not concern the mere elimination of the symptom and, at times, not even the treatment of the disorder.

 Thank you for the excellent remark! We addressed this in the discussion part.

---

## [Editor Report · Decision Letter 1]

21 Aug 2020

Digitally engaged physicians about the digital health transition

PONE-D-20-11916R1

Dear Dr. Győrffy,

We’re pleased to inform you that your manuscript has been judged scientifically suitable for publication and will be formally accepted for publication once it meets all outstanding technical requirements.

Kind regards,

Stefano Triberti, Ph.D.

Academic Editor

PLOS ONE

Additional Editor Comments (optional):

The manuscript has been adequately revised. Make sure to correct reference numbering according to the new references added. Moreover, Acknowledgment section thanks "e-patients", but this study was on physicians. Maybe Authors would like to correct this. 
---

## [Editor Report · Acceptance letter]

7 Sep 2020

PONE-D-20-11916R1

Digitally engaged physicians about the digital health transition

Dear Dr. Győrffy:

I'm pleased to inform you that your manuscript has been deemed suitable for publication in PLOS ONE. Congratulations! Your manuscript is now with our production department.

Kind regards,

on behalf of

Dr. Stefano Triberti 

Academic Editor

PLOS ONE